# Efficient Estimates of Surface Diffusion Parameters for Spatio-Temporally Resolved Virus Replication Dynamics

**DOI:** 10.3390/ijms25052993

**Published:** 2024-03-05

**Authors:** Markus M. Knodel, Gabriel Wittum, Jürgen Vollmer

**Affiliations:** 1Simulation in Technology, TechSim, 75248 Ölbronn-Dürrn, Germany; 2Modelling and Simulation (MaS), Computer, Electrical and Mathematical Science and Engineering (CEMSE), King Abdullah University of Science and Technology (KAUST), Thuwal 23955-6900, Saudi Arabia; gabriel.wittum@kaust.edu.sa; 3Institute for Theoretical Physics, Leipzig University, 04081 Leipzig, Germany; juergen.vollmer@uni-leipzig.de

**Keywords:** physical virology, hepatitis C virus (HCV), viral dynamics, within-host viral modeling, parameter estimation, 3D spatio-temporally resolved mathematical models, (surface) partial differential equations, realistic geometries, scaling analysis, geometry influence

## Abstract

Advanced methods of treatment are needed to fight the threats of virus-transmitted diseases and pandemics. Often, they are based on an improved biophysical understanding of virus replication strategies and processes in their host cells. For instance, an essential component of the replication of the hepatitis C virus (HCV) proceeds under the influence of nonstructural HCV proteins (NSPs) that are anchored to the endoplasmatic reticulum (ER), such as the NS5A protein. The diffusion of NSPs has been studied by in vitro fluorescence recovery after photobleaching (FRAP) experiments. The diffusive evolution of the concentration field of NSPs on the ER can be described by means of surface partial differential equations (sufPDEs). Previous work estimated the diffusion coefficient of the NS5A protein by minimizing the discrepancy between an extended set of sufPDE simulations and experimental FRAP time-series data. Here, we provide a scaling analysis of the sufPDEs that describe the diffusive evolution of the concentration field of NSPs on the ER. This analysis provides an estimate of the diffusion coefficient that is based only on the ratio of the membrane surface area in the FRAP region to its contour length. The quality of this estimate is explored by a comparison to numerical solutions of the sufPDE for a flat geometry and for ten different 3D embedded 2D ER grids that are derived from fluorescence z-stack data of the ER. Finally, we apply the new data analysis to the experimental FRAP time-series data analyzed in our previous paper, and we discuss the opportunities of the new approach.

## 1. Introduction

Research on virus infections is eminently important because virus infections pose major challenges to animal and human health and consequently impact global prosperity, the economy, and political and social systems [1,2,3,4,5,6]. Moreover, insights into the biological foundations of virus infections has revolutionized medical research by providing CRISPR/Cas [7,8] and facilitating genetic engineering [9,10,11,12]. For instance, this dichotomy is reflected in the cancer risk induced by persistent viruses [13,14,15], while insight into their strategies to avoid detection by the immune system helps unravel the workings of the human immune system [16] and spawns novel approaches to cancer therapy [17,18,19]. In addition to concealed long-time action, viruses also trigger epidemics and pandemics [20,21,22,23], such as the recent COVID-19 pandemic [24].

The present study addresses intracellular virus replication dynamics. Spatial dependence is a crucial factor in this process [24,25,26,27,28,29,30,31,32,33,34,35,36,37,38,39,40,41,42]. A lot of viruses cause the rearrangement of cellular membranes, and/or they induce the growth of specific regions within the host cell, and/or around cellular membranes. For example, the genetic information of some viruses gets replicated inside such regions in order to prevent immune system mechanisms that strive to stop virus genome replication. Here, we build on a framework of virus replication models for the hepatitis C virus (HCV) that aims at full 3D spatio-temporal resolution on an intracellular level [43,44,45,46,47]. HCV belongs to the family of plus-stranded RNA (ss(+)RNA) viruses [27,48], such as Dengue and yellow fever viruses (DNFV and YFV) [27,32], and Coronaviridiae such as SARS-CoV-1 and SARS-CoV-2 [24]. The aim of this project is to develop an “in silico microscope” for direct comparisons between experiments and simulations of biophysical models of intracellular virus replication [49,50,51]. The present approach augments this method by estimates of material parameters derived directly from the experimental data.

The nonstructural proteins (NSPs) of HCV govern the replication of the HCV genome (viral RNA—vRNA) [27,52]. Directly after their cleavage from the viral polyprotein, which is translated at the ribosomes from the information stored within the vRNA attached to the ribosomes, the NSPs anchor to the ER surface [27,28,30,48,53,54]. Thus, the movement of the NSPs is restricted to the ER surface, which is a 2D curved manifold embedded in 3D. Refs. [43,44,45,46,47] described the evolution of the vRNA cycle by surface partial differential equations (sufPDEs). These equations were evaluated numerically on reconstructed geometry grids describing the ER and membranous web (MW) regions.

To account for the requirement that parameters are valid for quantitative biophysical model simulations, a first estimation of the diffusion constant of the HCV NS5A [29,31,54] NSP on the ER surface was established in [45] based on the Bayesian averaging of a considerable number of single numeric sufPDE computations [43]: In vitro fluorescence recovery after photobleaching (FRAP) experiment [55,56] time series (TMS) [53] data were fitted with sufPDE computations [43], with each performed at several ER surface geometries reconstructed from data [25] to account for the unknown FRAP experiment cell geometry. This approach assumed that the NS5A diffusion coefficient varies slightly in each FRAP experiment and for each cell geometry, thus resulting in a substantial amount of computing time [46].

In the present paper, we provide a scaling analysis [57,58] of the solutions of the sufPDEs. We explore how the recovery of bleaching is impacted by the surface geometry, and we establish how the surface diffusivity can be inferred based on a scaling plot. This approach assumes that the diffusion coefficient is the same for all experiments with the same cell type, and that differences in FRAP time series are exclusively due to different shapes of the ER. This suggests an algorithm to determine the surface diffusivity that relies solely on the knowledge of the initial fluorescence and a length scale characterizing the geometry of the ER. These conclusions are underpinned by simulations that account for the complex geometric and topological structures of ERs. Finally, the application of the new data analysis method is demonstrated by revisiting the FRAP time-series data analyzed in [45,53].

This paper is organized as follows: In Section 2.1, we revisit the biological problem, and we present our mathematical representation of the experimental measurements. Section 2.2 provides a thorough scaling analysis of the model equations. The established scaling is our main result. In Section 3, we demonstrate that scaling facilitates a novel approach to analyzing FRAP data. The application of the method is demonstrated for data described and analyzed in Ref. [45]: we determine the diffusivities of the surface proteins, compare the results to the values obtained previously, and showcase the benefits of the new approach. The methods involved in this comparison are documented in Section 4. Finally, in Section 5, we discuss and summarize our findings.

## 2. Results

The present paper addresses the analysis of in vitro fluorescence recovery after photobleaching (FRAP) [55,56]. The measurements provide direct access to the transport properties of fluorescently labeled substances in the cell.

### 2.1. FRAP Experiments and Their Mathematical Modeling

As concrete example, we consider FRAP experiments [53] where NS5A (a major HCV NSP [26,29,31,52,54,59]) proteins are marked by a fluorescent dye. The proteins reside on the ER surface in liver cells. The cell is prepared in a state where the protein is distributed (and fluoresces) homogeneously. In Figure 1, the ER appears as a bicontinuous structure colored in red and blue. The ER has a height of a few pore diameters and large lateral extension. The experiment starts by applying a bright laser pulse applied to a small and sharply confined spatial region. In Figure 1, this region is colored in blue. The pulse bleaches the dye and hence quenches the fluorescence in this region. In the following, these regions will be called the FRAP region of interest (FRAP ROI) F. This region has the shape of a cylinder, with a height that amounts to the thickness of the ER, H≃3.5 μm, and a circular cross-section of $A=\pi \;R^{2}$ with R≃3.5 μm.

After bleaching, the diffusive rearrangement of NS5A between the region and its environment is recorded by tracking the recovery of fluorescence. Fluorescence recovers due to the influx of NS5A proteins labeled with operational dye molecules from the surrounding unbleached region U. The recovery is followed by the induction of dye fluorescence by the repetitive application of a second (soft) laser beam. Representative measurement signals are shown by the data points in Figure 2. The initial recovery up to a level of 0.35 proceeds within a few seconds. Subsequently, the experimental signal keeps rising slowly. However, even the soft laser beam causes photobleaching, and this photobleaching slowly extinguishes the fluorescence signal. The full details are provided in Ref. [45].

We express the dynamics of NS5A fluorescence in terms of the intensity density,
$$\begin{array}{c} i_{\mathrm{ns}5a}\left(t,x\right)\propto c_{\mathrm{ns}5a}\left(t,x\right) \end{array}$$
which is proportional to the concentration $c_{\mathrm{ns}5a}\left(t,x\right)$ anchored in the ER. We consider five ER geometries that are reconstructed based on experimental data (cf. Section 4). The evolution of the protein on this ER surface domain E is described by a surface PDE (sufPDE): (1)$$\begin{array}{c} \frac{d}{dt}i_{\mathrm{ns}5a}\left(t,x\right)-D\Delta_{\left(T\right)}i_{\mathrm{ns}5a}\left(t,x\right)=-{\left.r_{p}\;i_{\mathrm{ns}5a}\right|}_{F}\;,\;\forall x\in E=F\cup U. \end{array}$$
where the Laplace–Beltrami operator $\Delta_{\left(T\right)}$ is the projection of the Laplace operator to the tangential space of the two-dimensional ER-hypersurface E=F∪U. It models diffusion analogously to a Laplace operator for diffusion in Euclidean space. The right-hand side of the equation accounts for the intensity reduction due to the quenching of the dye in the course of the intensity measurement with the soft laser.

The present analysis adopted simulations without quenching $r_{p}=0\;s^{-1}$ and with a quench rate $r_{p}≃0.0020\;s^{-1}$, which is slightly larger than the values $0.0011\;s^{-1}$ and $0.0016\;s^{-1}$ reported for NS5A/Alone and NS5A/OtherNSPs cells, respectively [45]. The surface diffusion coefficient *D* varies in the range of $5\times 10^{-4}$, …, $1\times 10^{-1}\;\mu m^{2}/s$.

In order to match the experimental setup, the initial conditions for Equation (Equation 1) are provided independently for the bleached (F) and unbleached (U) regions: (2)$$i_{\mathrm{ns}5a}\left(t_{0},x\right)=\left\{\begin{array}{cc} i_{0}^{F}\;, & \forall x\in F\;,\\ i_{0}^{U}\;, & \forall x\in U\;. \end{array}\right.$$For a comparison with the experimental intensity in F, we define the integrated normalized luminosity: (3)$$I\left(t\right)={\langle i_{\mathrm{ns}5a}\left(t,x\right)\rangle }_{F}=\frac{{\;\int \int \;}_{F}dS\;i_{\mathrm{ns}5a}\left(t,x\right)}{{\;\int \int \;}_{F}dS}$$
where dS denotes surface area elements on the ER. The time evolution of I(t) is shown in Figure 2.

Figure 1 displays a screenshot of a simulation movie for the NS5A dynamics on the ER surface mimicking a FRAP experiment. The simulation movie itself is attached as a Appendix A to this study.

### 2.2. Scaling Analysis of FRAP ROI Intensity

In order to discuss the solutions of Equation (Equation 1), we introduce the surface area in the hole,
$$\begin{array}{cc} \left|F\right| & =\underset{F}{\;\int \int \;}dS\;. \end{array}$$
and the space-dependent absorption rate
$$\begin{array}{c} \gamma \left(x\right)=r_{p}\;\chi \left(\ell \left(x\right)\right) \end{array}$$
where ℓ(x) is the signed distance of x to the boundary ∂F of F (with negative values denoting positions inside F), and
$$\begin{array}{c} \chi \left(\ell \right)=\left\{\begin{array}{cc} 1 & \;\;\mathrm{for}\;\;\;\;\ell \leq \;0\\ 0 & \;\;\mathrm{for}\;\;\;\;\ell >0 \end{array}\right. \end{array}$$With these, we write Equation (Equation 1) as
(4)$$\begin{array}{c} e^{-\gamma (x)\;t}\frac{d}{dt}\left(e^{\gamma (x)\;t}\;i_{\mathrm{ns}5a}\left(t,x\right)\right)=D\;\Delta_{\left(T\right)}i_{\mathrm{ns}5a}\left(t,x\right) \end{array}$$Multiplying by $\exp\left(\gamma (x)\;t\right)$, dividing by A=|F|, and integrating both parts over F provide
(5)$$\begin{array}{cc} \frac{d}{dt}\left(e^{r_{p}\;t}\;I\left(t\right)\right) & =\frac{D}{\left|F\right|}\underset{F}{\;\int \int \;}dS\;e^{\gamma (x)\;t}\;\Delta_{\left(T\right)}i_{\mathrm{ns}5a}\left(t,x\right)\\ \; & =\frac{D}{\left|F\right|}\;\;e^{r_{p}t}\;\underset{F}{\;\int \int \;}dS\;\nabla_{\left(T\right)}\;\nabla_{\left(T\right)}i_{\mathrm{ns}5a}\left(t,x\right)\\ \; & =\frac{D}{\left|F\right|}\;\;e^{r_{p}t}\;\oint_{\partial F}d\hat{n}\;\cdot \;\nabla_{\left(T\right)}i_{\mathrm{ns}5a}\left(t,x\right) \end{array}$$Here, we first used the fact that $\gamma \left(x\right)=r_{p}$ takes a constant value on F, and then we adopted Gauss’s theorem to express the integral over F as an integral over its surface ∂F.

Equation (Equation 5) must be solved for the initial condition
$$\begin{array}{c} I\left(0\right)={\langle i_{\mathrm{ns}5a}\left(t_{0},x\right)\rangle }_{\;\;F}=i_{0}^{F}\;. \end{array}$$

#### 2.2.1. Scaling for Negligible Photobleaching

In the absence of photobleaching, $r_{p}≃0$, Equation (Equation 5) reduces to
(6a)R2DddtI(t)=R2|F|∮∂Fdn^·∇^(T)ins5a(t,x),(6b)withR2Dddtins5a(t,x)=R2Δ(T)ins5a(t,x).We non-dimensionalize the equations by adopting the radius *R* of the FRAP region as the length scale and the dimensionless time $\tau =D\left(t-t_{0}\right)/R^{2}$. Moreover, we note that the Equations (6) are linear (integro-)differential equations. As a consequence, there is a superposition principle, and one can account for the different values of $i_{0}^{U}$ and $i_{0}^{F}$ in the initial condition by considering the normalized intensity $\tilde{I}\left(\tau \right)=\left(I\left(t\right)-I_{0}\right)/\left(\left|i_{0}^{U}-i_{0}^{F}\right|\right)$ (see the chapter on Greens functions in Ref. [60]). This has two consequences: 1. Initially, the density $i_{0}^{F}$ in F is uniform such that it agrees with its average value $I_{0}$. 2. The reduced intensity undergoes the same evolution for every geometry of the ER, irrespective of the value of the diffusion coefficient and the initial condition.

This prediction is tested in Figure 3, where we plot data for the different geometries of the ER and for the 2D planar geometry. For each geometry, the data for different diffusion coefficients collapse on a master curve. However, the data for different geometries lie on distinctly different curves. The values of data for different ER geometries vary by about 20%, and those for the disk geometry (uppermost curve in the plot) take noticeably higher values. This highlights the importance of the ER geometry for FRAP relaxation.

Figure 3 suggests that the relaxation of the reduced density has a time dependence proportional to the square root of the dimensionless time. We will show now that this is an immediate consequence of the form of the initial condition.

For the initial condition, Equation (Equation 2), the intensity $i_{\mathrm{ns}5a}\left(t,x\right)$ takes the form of the step function,
$$\begin{array}{c} i_{\mathrm{ns}5a}\left(t_{0},x\right)=\frac{i_{0}^{U}+i_{0}^{F}}{2}-\chi \left(\ell (x)\right)\;\frac{i_{0}^{U}-i_{0}^{F}}{2}\;. \end{array}$$For early times, the flow is orthogonal to the surface; i.e., the curvature and topology of the surface may still be neglected. The gradient of the density therefore follows the evolution of the diffusive decay of a step function in one dimension, i.e., of the derivative of a delta function. Consequently, the time evolution of the reduced density is obtained as the derivative of the Greens function of diffusion. This amounts to the derivative of a Gaussian function [60]: (7)$$\begin{array}{c} \nabla_{\left(T\right)}i_{\mathrm{ns}5a}\left(t_{0},x\right)=\frac{i_{0}^{U}-i_{0}^{F}}{2\sqrt{4\pi \;D\;t}}\;\exp\left(-\frac{\ell^{2}}{4D\;t}\right) \end{array}$$Right at the interface, we have ℓ=0, and the exponential function takes a value of one. Hence, Equation ([Disp-formula FD6a-ijms-25-02993]) reduces to
(8)$$\begin{array}{c} \frac{\Lambda^{2}}{D}\frac{d}{dt}I\left(t\right)=\frac{i_{0}^{U}-i_{0}^{F}}{2}\sqrt{\frac{\Lambda^{2}}{D\;t}}\;\mathrm{with}\;\Lambda =\sqrt{4\pi }\;\frac{\left|F\right|}{\left|\partial F\right|} \end{array}$$Integrating time from $t_{0}=0$ until *t* provides
(9)$$\begin{array}{c} \frac{I\left(t\right)-I_{0}}{i_{0}^{U}-i_{0}^{F}}=\sqrt{\frac{D\;t}{\Lambda^{2}}} \end{array}$$This result explains the square-root dependence of the intensity observed in Figure 3, and it provides the leading-order dependence of the dependence on the surface geometry via the characteristic length scale $\Lambda =\sqrt{4\pi }\;|F|/\left|\partial F\right|$.

The prediction is tested in Figure 4a. In the late stage of the recovery the recovery is still considerably impacted by the ER geometry. The curves for different geometries differ by up to 25%. However, for the data at the outset the transformation from a time scale $Dt/R^{2}$ to $Dt/\Lambda^{2}$ provides a very nice data collapse for the ER geometries. For a perfect fit, the ratio of the data to $\sqrt{D\;t/\Lambda^{2}}$ must be one. For our simulations, it takes a value close to $\phi_{-}=1.25$. We attribute this difference to the impact of the nontrivial curvature of the border of F: the solution Equation (Equation 7) applies when the normal direction $\hat{n}$ at the interface of F does not change when moving along the interface. This holds for a straight line in a plane and for a vertical section through a cylinder. For the surface of the disks, this condition is always violated, such that the prefactor takes a value substantially larger than one. For the surface of F, it applies to a good approximation at some place. Hence, the prefactor takes a value closer to one than that for a disk.

#### 2.2.2. Accounting for Photobleaching

The data collapse at early times also holds for systems with bleaching (Figure 4b). However, at late times, these systems cross over to a decay of the intensity that is caused by bleaching. In order to gain insight into this behavior, we insert Equation (Equation 7) into Equation (Equation 5),
(10)$$\begin{array}{c} \frac{1}{r_{p}}\;\frac{d}{dt}\left(e^{r_{p}\;t}\;I\left(t\right)\right)=\frac{K}{\sqrt{r_{p}t}}\;e^{r_{p}\;t}\;\mathrm{with}\;K=\sqrt{\frac{D}{\Lambda^{2}\;r_{p}}}\;\frac{i_{0}^{U}-i_{0}^{F}}{2} \end{array}$$
where *K* is a dimensionless constant. This equation can be integrated
$$\begin{array}{cc} e^{r_{p}\;t}\;I\left(t\right)-I_{0} & =K\;\int_{0}^{r_{p}t}d\tau \;\frac{e^{\tau }}{\sqrt{\tau }}=K\;\int_{0}^{i\;\sqrt{r_{p}t}}d\theta \;\left(-2i\right)\;e^{-\theta^{2}}=-K\;i\;\sqrt{\pi }\;\mathrm{erf}\left(i\;\sqrt{r_{p}t}\right) \end{array}$$
where erf(x) is the error function. Rearranging the terms provides
(11)$$\begin{array}{cc} \frac{I\left(t\right)-I_{0}\;e^{-r_{p}\;t}}{K} & =+\frac{\sqrt{\pi }}{i}\;\mathrm{erf}\left(i\;\sqrt{r_{p}t}\right)\;e^{-r_{p}t} \end{array}$$This prediction is tested in Figure 5a. Indeed, we observe a data collapse. It features a cross-over between different power laws for small and large values of $r_{p}t$.

For $r_{p}t\ll 1$, the exponential function and the error function on the right-hand side can be expanded in a power series. This entails a square-root dependence for early times:
(12a)$$\begin{array}{c} \frac{I\left(t\right)-I_{0}\;e^{-r_{p}t}}{K}≃2\;\sqrt{r_{p}\;t}\;\left(1-\frac{2}{3}\;r_{p}t+O{\left(r_{p}t\right)}^{2}\right) \end{array}$$This is in line with Figure 5a, where the data for small $r_{p}t$ lie close to $2\;\phi_{-}\sqrt{r_{p}t}$, with $\phi_{-}=1.25$. Here, the correction factor $\phi_{-}$ accounts for the curvature effects reported in Figure 4a.

To find the asymptotics for $r_{p}t\gg 1$, we observe that
(12b)$$\begin{array}{cc} \frac{I\left(t\right)-I_{0}\;e^{-r_{p}t}}{K} & =\int_{0}^{r_{p}t}d\tau \;\frac{e^{\tau -r_{p}t}}{\sqrt{\tau }}=\int_{0}^{r_{p}t}d\tau \;\frac{e^{-\tau }}{\sqrt{r_{p}t-\tau }}=\frac{1}{\sqrt{r_{p}t}}\;\left(1+\frac{1}{2r_{p}t}+\;O{\left(r_{p}t\right)}^{-2}\;\right) \end{array}$$This is in line with the data collapse in Figure 5a, where the data for large $r_{p}t$ lie close to $2\;\phi_{+}/\;\;\sqrt{r_{p}t}$ with $\phi_{+}=2$. Here, a correction term, $\phi_{+}$, of the prefactor is expected since Equation (Equation 7) is based on an approximation that applies only to early times. An extrapolation that agrees very well with the observed data is provided by
(13)$$\begin{array}{cc} \frac{I\left(t\right)-I_{0}\;e^{-r_{p}t}}{K} & ≃\frac{2\;\phi_{-}\;\phi_{+}\;\sqrt{r_{p}t}}{\phi_{+}+\phi_{-}\;r_{p}t}\;. \end{array}$$It is shown by the thick solid gray line in Figure 5.

## 3. Discussion

The analysis in Section 2.2 suggests two approaches to the analysis of FRAP data. We explore the chances and challenges by revisiting the analysis of the data introduced in Figure 2. The best fits of the analysis in Ref. [45] and those that will be derived in the following are summarized in Table 1.

### 3.1. Analysis Based on Equation (Equation 9)

The first option is to collect a vast amount of data for times where bleaching still has a negligible impact. To work out this condition, we expand the exponential function on the left-hand side of Equation ([Disp-formula FD12a-ijms-25-02993]), insert the expression of *K*, and rearrange the terms. This provides Equation (Equation 9) and its leading-order corrections:(14)$$\begin{array}{cc} \frac{I\left(t\right)-I_{0}}{i_{0}^{U}-i_{0}^{F}} & ≃\sqrt{\frac{D\;t}{\Lambda^{2}}}\;\left(1-\frac{2}{3}\;r_{p}t+O{\left(r_{p}t\right)}^{2}\right)-\frac{I_{0}\;r_{p}t}{i_{0}^{U}-I_{0}}\;\left(1+O(r_{p}t)\right)\\ \; & =\sqrt{\frac{D\;t}{\Lambda^{2}}}\;\left(1-\frac{I_{0}}{i_{0}^{U}-I_{0}}\;\sqrt{\frac{r_{p}\;\Lambda^{2}}{D}}\;\sqrt{r_{p}t}-\frac{2}{3}\;r_{p}t+O{\left(r_{p}t\right)}^{3/2}\right) \end{array}$$In Figure 6a,b, we plot the left-hand side of Equation (Equation 14) as a function of *t*. The two panels refer to the two cell lines introduced in Figure 2. The lines with different colors show the results of repeated experiments. In view of Figure 4a, we expect a square-root dependence, $\phi_{-}\;\sqrt{D/\Lambda^{2}}\;\sqrt{t}$. Hence, one can infer *D* based only on the knowledge of the geometrical parameter Λ of the measurement region—provided that one clearly resolves the power law at early times. For the first cell line, this is given. For the second cell line, we do not recommend this analysis. However, for the provision of the reader, we provide the power laws for the values obtained in [45] and those that will be derived in Section 3.2 by solid black and gray lines, respectively. A direct fit of the data (if successful) provides a value of *D* somewhere in between these two values (cf. Table 1).

Equation (Equation 14) entails that the corrections are small as long as
$$\begin{array}{c} r_{p}t\ll \min\left(\frac{3}{2}\;;\;\;\frac{D}{r_{p}\Lambda^{2}}\;{\left(\frac{i_{0}^{U}}{I_{0}}-1\right)}^{2}\right) \end{array}$$For the values of $r_{p}$ and *D* in [45], the latter factors take the values 2.5 and 0.5, respectively. The fit should be performed for times earlier than 50 s and 15 s, respectively, when one requires that the corrections be smaller than 5% of the leading contribution. This entails that the impact of photobleaching in the second experiment can be considerably reduced by performing experiments with stronger initial photobleaching, i.e., a smaller $I_{0}$.

In summary, this approach to data analysis can be applied when the range of data up to the cutoff time spans at least one order of magnitude. The estimate of *D* will then only rely on knowledge of the expectation of the parameter Λ, i.e., on a geometric parameter characterizing the geometry of the ER in F. When |F| and |∂F| are measured for some structures, it can most reliably be obtained by inspecting the cumulative distribution function of $\Lambda =\sqrt{4\pi }\;\left|F\right|/\left|\partial F\right|$, as shown in Figure 5b.

### 3.2. Analysis Based on Equation (Equation 11)

Equation (Equation 11) provides a slightly more involved data analysis that takes into account the impact of photobleaching during the experiments. It is based on the observation in Section 2.2.2 that the data collapse of Equation (Equation 11) can be described as a cross-over of two power laws, as in Equation (Equation 13) (cf. Figure 5), and this interpolation entails that
(15)$$\begin{array}{cc} I(t) & ≃I_{0}\;e^{-r_{p}\;t}+K\;\frac{2\;\phi_{-}\;\phi_{+}\;\sqrt{r_{p}t}}{\phi_{+}+\phi_{-}\;r_{p}t} \end{array}$$Hence, for every given experimental data set, we obtain *K* for a given $r_{p}$ by fitting I(t) with the right-hand side of Equation (Equation 15). In view of Equations (Equation 8) and (Equation 10), the variability in the values of *K* reflects to a substantial extent the different structures of the ER in the measurement region. This is reflected in the substantially improved data collapse shown in Figure 6. In these plots, we determine the preferred choice of $r_{p}$ by looking for the best match of the fit (solid gray line) and the expectation of the data for the largest values of $r_{p}t$. In the present case, this amounts to $r_{p}=0.004\left(1\right)\;s^{-1}$ for both data sets. This reflects the expectation that the quench rate is independent of the investigated cell line. It is a parameter set by the properties of the dye and the laser system in the setup. Accordingly, we find the best fits $D=0.075\left(15\right)\mu m^{2}/s$ and $D=0.034\left(13\right)\mu m^{2}/s$ for the data shown in Figure 7a,b, respectively. These regions are marked by green rectangles in Figure 7c,d. The estimates for other quench rates are given by plus symbols with error bars in those panels.

Moreover, in this figure, the estimates suggested in Ref. [45] are shown by large squares. These estimates were obtained, however, for smaller values of $r_{p}$ because the analysis in Ref. [45] is biased to provide quench rates that are systematically too small. After all, it neglects the diffusion of dye molecules into the test region considered in the estimate of $r_{p}$. On the other hand, Equation (A1) and Table A5 of Ref. [45] provide predictions of how the values of the diffusion coefficients change when adopting other values of $r_{p}$. It is shown by the solid lines shown in Figure 7c,d. For $r_{p}=0.004\;s^{-1}$, we find $D=0.061\;\mu m^{2}/s$ and $D=0.018\;\mu m^{2}/s$ for the first and second cell lines, respectively. These values lie on the smaller side but are still within the error margins of the present analysis. They are shown by the small squares.

We believe that the present approach to the data analysis provides a more reliable estimate of $r_{p}$, which is based on the bending of the lines shown in Figure 7a,b. Moreover, it explicitly accounts for the substantial correlations between the structure of the ER in the measurement region and the observed FRAP signal, while [45] adopted a Bayesian average of the different diffusion coefficients encountered when aiming for the best match of the relaxation of measurement data and simulations performed for a different ER geometry. We believe that this is the origin of the slight difference of the present estimate and the values suggested in Ref. [45].

## 4. Materials and Methods

Numerical tests of our theoretical considerations were performed by numerically solving Equation (Equation 1) on reconstructions of ER geometries that are fully described in [43,45,46]. An example is provided in Figure 8. The ER [25,61,62] comprises a bicontinuous structure with a characteristic pore diameter of 1 μm. It has a thickness of a few pore diameters and a lateral extension of about 50 pore diameters in the other two directions. The overall surface area of the membrane comprising the ER is on the order of $\left|E\right|≃5000\;\mu m^{2}$. We considered five surface geometries, denoted by M1 to M5. For each of them, we considered two distinct locations of the FRAP region of interest. This distinction is marked by *a* and *b*, respectively. Hence, for the simulations, we considered ten ER structures in total. Moreover, in order to explore the impact of the bicontinuous structure of the ER, we also performed simulations on a flat disk. This structure is denoted as a disk. The specific values for the structures considered in the present study are provided in the table in Figure 8. Full details concerning the investigated structures and the numerical method are provided in [43,45,46].

The ER geometries are represented by subdomains of a computational mesh that is derived from fluorescence stacks of stained cell structures. The surfaces are represented by triangulated manifolds with a resolution of about $10^{6}$ nodes [45]. Figure 8 shows an example of such a geometric setup.

In Ref. [45], the diffusion constant of the NS5A protein was estimated based on 20 time series (TMS) of fluorescence recovery after photobleaching (FRAP) experiments [53,55,56]. For each of these time series, the diffusion coefficient was estimated by obtaining an optimal match with the numerical solutions of the sufPDE for each of 10 different ER geometries [25].

The authors of Ref. [45] adopted a Gauss–Newton recursion scheme to determine the value of the diffusion constant *D* that provided the best match between the numerically computed intensity and a FRAP scan covering 125 s of experimental data. Typically, around 50 simulation runs were required to reach convergence. Each of these had a resolution of about 1 Million degrees of freedom [46] at the base level. The value of the surface diffusion coefficient was estimated from these data by minimizing the mismatch over all combinations of experimental data and numerical time series [45].

This procedure was chosen because the geometry of the ER is not recorded in FRAP experiments [53]. Finding the optimal matching by the Gauss–Newton procedure involved about $4\times 10^{4}$ simulations with different sets of geometries and material parameters and about 4 Million core hours at the HLRS Stuttgart hazel hen supercomputer [46]. In the end, the global optimum was obtained as a Bayesian average [45].

The present sufPDE simulations were performed with the simulation software UG4 (https://github.com/UG4, accessed on 27 February 2024) [63,64] at the Apollo Hawk supercomputer of the HLRS Stuttgart, Baden-Württemberg, Germany.

## 5. Conclusions

We revisited a large body of highly resolved numerical data addressing the diffusion of nonstructural HCV proteins (NSPs) that are anchored in the endoplasmatic reticulum (ER). For the considered system, we have access to structural information about the ER (Figure 1 and Figure 8) and to FRAP measurements (Figure 2). We established scaling in the transport equations governing the diffusion process, and we inspected the resulting data collapse for all available numerical data (Figure 5). The data fall on a master curve that is described to a very good approximation by Equation (Equation 15).

Our analysis reveals that the geometrical structure and the topology of the ER in the FRAP measurement area have a severe impact on the evolution of the FRAP signal. Based on these insights, we suggest a novel approach to analyzing experimental data that is outlined in Section 3.2. This analysis is based on fitting experimental data, and it only requires information about the expectation value of a single parameter, Λ, that characterizes the geometry of the FRAP measurement region: the parameter Λ amounts to the ratio of the surface area in the measurement region to the length of its boundary (Equation (Equation 8)).

A very accurate quantitative evaluation of the diffusion constant is obtained in Figure 7 by introducing corrections factors, $\phi_{\pm }$, that account for curvature and quenching effects. These factors take values close to one. In the present study, they have been inferred by inspecting highly resolved numerical data of diffusive recovery for the diffusion on two-dimensional bicontinuous surfaces extracted from experimental measurements of the endoplasmatic reticulum (ER).

The present approach applies to the analysis of all FRAP signals of proteins that are anchored to the endoplasmatic reticulum or other biological structures with a complex two-dimensional shape. In a first approximation, the fit function, Equation (Equation 13), can be adopted for the fits. The value of Λ appears in the fits through the definition of *K*, Equation (Equation 10). The value of this ratio can be inferred from measurements of the geometry of the ER by fluorescence imaging [25] or electrotomography [26]. The values of $\phi_{\pm }$ will likely not change a lot. However, for an accurate data analysis, it will be worthwhile to perform a numerical study of new systems to adjust these empirical numerical factors in the fit function.

With this precaution, one establishes a straightforward method to determine the diffusion parameters for viral proteins that are attached to the ER surface. Surface diffusivity of NSPs on the ER is an important material parameter that is needed to arrive at quantitative predictions of the rates of the biochemical reaction kinetics of processes located on the ER [53,55,56,65,66]. For instance, we recently adopted the NSP surface diffusion constant derived in the present study in a coupled surface–volume PDE model [50,51]. This allowed us to perform biophysical simulations for the dynamics of the major components of an intracellular virus genome replication cycle which are in quantitative agreement with experimental data [49].

In conclusion, the new data analysis can also be used to obtain the diffusion coefficients of arbitrary cell components that are located on the ER surface, such as specific metabolic proteins. These transport parameters are essential inputs to describe all diffusion-limited biochemical reactions rates in spatially heterogeneous cell environments [65,66,67,68], particularly when the reactions take place on two-dimensional membranes [69].

## Figures and Tables

**Figure 1 ijms-25-02993-f001:**
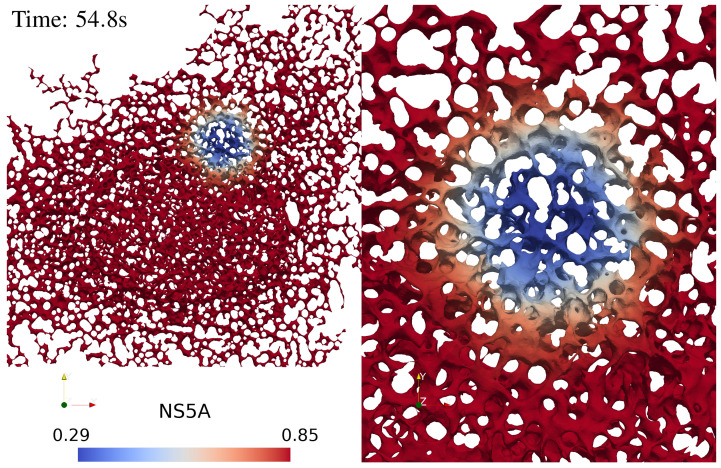
A screenshot of the evaluation of NS5A FRAP experiment simulation based on the sufPDE description, Equation (Equation 1), on an ER surface: (**left**) the total view, (**right**) a zoom into the FRAP ROI event. The color coding indicates the density of dyed surface proteins embedded in the ER. Its gradients decay in the course of the FRAP measurement. A movie of this evolution is attached as Appendix A to the present study.

**Figure 2 ijms-25-02993-f002:**
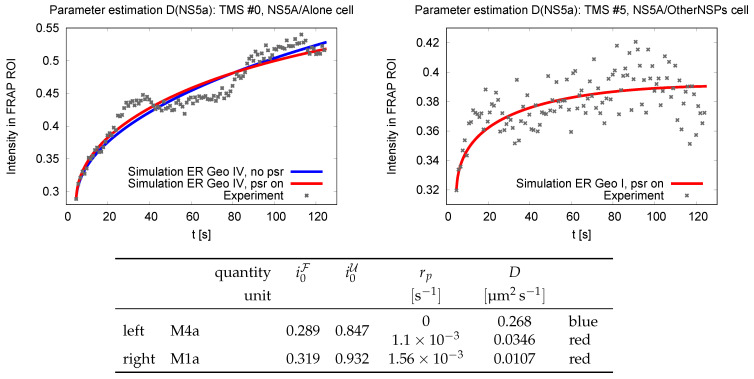
Experimental (dots) and simulation (solid lines) FRAP signals for two different cell lines, where one expects different surface diffusivities *D*. The blue and red lines show simulation results where photobleaching is neglected and for a finite quench rate, respectively. The geometry of the ER of the cells used in the experiments is not known. The simulations were performed for the geometries M4a (**left**) and M1a (**right**), respectively. The initial conditions and parameters adopted in the simulations are provided in the table below the plots. Full details are provided in Section 4. [This figure was initially published as Figuress 5 and 6 in Ref. [45]. The abbreviations “TMS # - time series number” were originally introduced in [45] and number the experimental FRAP time series [53] for which the diffusion constant was adjusted.

**Figure 3 ijms-25-02993-f003:**
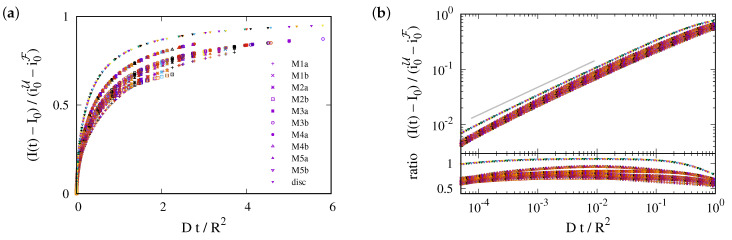
Data collapse for the reduced density $\left(I\left(t\right)-I\left(0\right)\right)/\left(\left|i_{0}^{U}-i_{0}^{F}\right|\right)$ as a function of the dimensionless time $D\;t/R^{2}$. Different colors refer to different values of *D*, which take values in $D\in \{0.5\times 10^{-3}\;\mu m^{2}/s\;,\;1.0\times 10^{-3}\;\mu m^{2}/s\;,\;\cdots \;,\;0.1\;\mu m^{2}/s\}$. The symbols refer to different geometries, as specified in the legend, and discussed in Section 4. (**a**) Simulation data with a linear scale. There are different curves for each geometry, but for any given geometry, data with different *D* collapse on a single line. (**b**) The **upper panel** provides a double-logarithmic plot of the data shown in (**a**). The **lower panel** shows the ratio of the data and $\sqrt{D\;t/R^{2}}$. To a good approximation, it is a horizontal line. Hence, the data follow a power law with the exponent 1/2, where the ratio provides the prefactor of the power law. For the planar geometry, the prefactor takes a value slightly larger than one. For bicontinuous structures, it differs for each geometry, with values in the range between 0.5 and 0.8.

**Figure 4 ijms-25-02993-f004:**
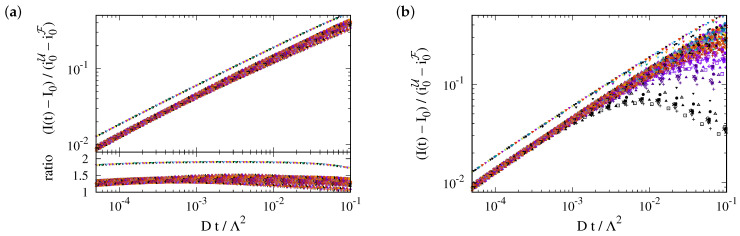
(**a**) The master plot, Equation (Equation 9), for the data shown in Figure 3b. The values of Λ depend on the geometry. This reduces the variance of the prefactors by roughly a factor of two. For the ER geometries, they take values of about $\phi_{-}=1.25\left(7\right)$. For a 2D disk, it is close to 1.8. (**b**) The same master plot for data with non-vanishing photobleaching, $r_{p}$. At early times, the data still follow the prediction, but now there is a cross-over to a regime where photobleaching has a severe impact.

**Figure 5 ijms-25-02993-f005:**
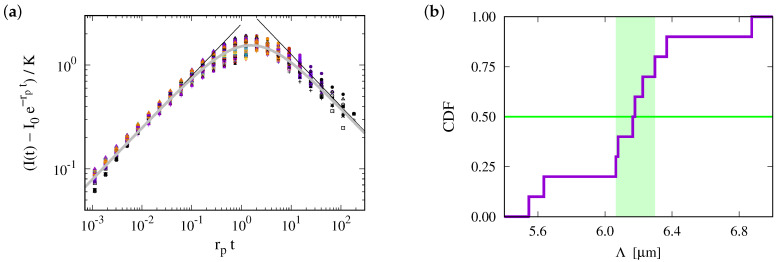
(**a**) The collapse of all data on the master curve predicted by Equation (Equation 11). There is a very good collapse for early times, $r_{p}\;t≲0.1$, and the cross-over due to photobleaching is also accounted for. The two straight gray lines show the functions $2\;\phi_{-}\;\sqrt{r_{p}t}$ and $2\;\phi_{+}/\;\;\sqrt{r_{p}t}$, with $\phi_{-}=1.25$ and $\phi_{+}=2$, respectively, i.e., the asymptotic power laws derived in Equation (12). The thick solid gray line shows an interpolation, Equation (Equation 13), of the asymptotics. (**b**) The cumulative distribution of the values of $\Lambda =\sqrt{4\pi }\left|F\right|/\left|\partial F\right|$. For F with a radius of R=3.5 μm in the presently considered ER geometries, the values of Λ are sharply centered in a narrow interval with a width of 0.3
μm (shaded region) around a median of $\bar{\Lambda }≃6.17\;\mu m$.

**Figure 6 ijms-25-02993-f006:**
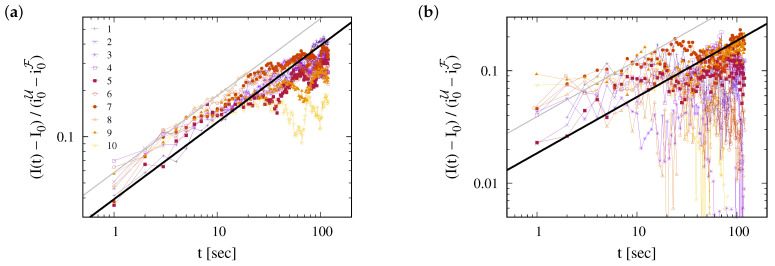
Fitting the diffusion constant to experimental data values. Panels (**a**,**b**) show data for the two cell lines introduced in Figure 2, where the left-hand side of Equation (Equation 14) is plotted as a function of time. The solid black lines show the prediction of Equation (Equation 9) with the correction factor $\phi_{-}=1.25$ when it is evaluated for the data obtained in Ref. [45]. The gray lines provide the dependencies by the best fits obtained for the values of *D* obtained in Section 3.2. See Table 1 for a summary of these values.

**Figure 7 ijms-25-02993-f007:**
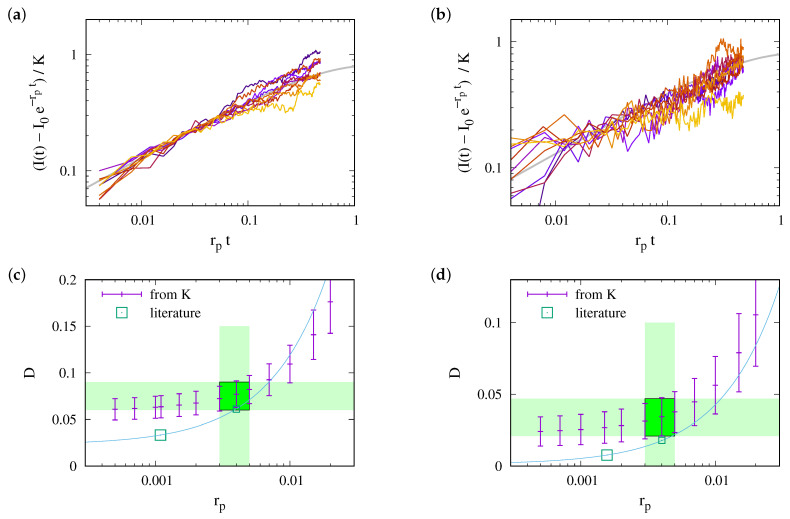
An analysis of the experimental data considered in Figure 6 that is based on Equation (Equation 15). Panels (**a**,**b**) show data for the two cell lines together with the best fits for $r_{p}=4\times 10^{-3}\;s^{-1}$, where the fit proceeds through the expectation of the distribution of data for large values of $r_{p}t$. Panels (**c**,**d**) show the prediction of *D* obtained by the analysis of the factor *K*. The small points with error bars refer to our present data. The points indicated by large boxes provide the values suggested in Ref. [45], and the solid line shows the extrapolation to other quench rates described in the appendix of that paper. For the quench rates obtained in the present analysis, the predictions agree within the error margins.

**Figure 8 ijms-25-02993-f008:**
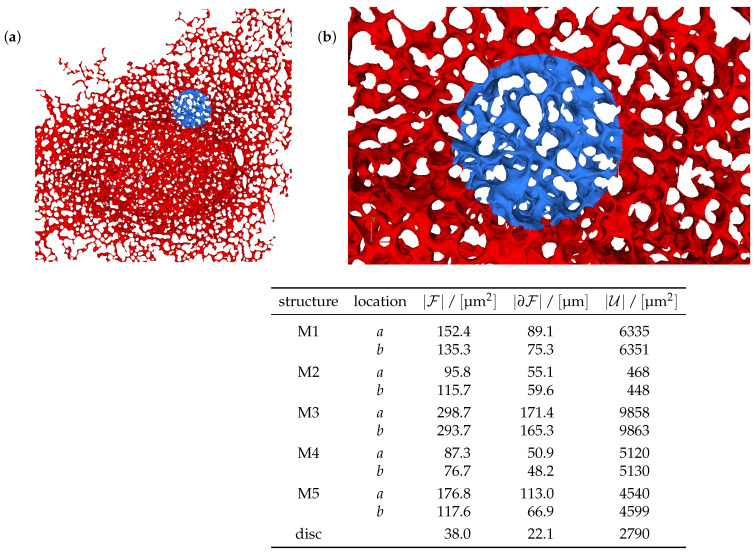
The surface mesh of a reconstructed ER geometry. (**a**) The computational domain used for the simulations of the FRAP experiments of NS5A. The displayed ER has a maximum thickness of about 4.8 μm, which amounts to only a few pore diameters, and the present view has a lateral extension of 46 μm×46 μm. The unbleached region U is displayed in dark red, and the FRAP region F used for bleaching is in blue. (**b**) Magnification around the FRAP region F with an extension of 19.2 μm×10.9 μm and a maximum thickness of about 3.5 μm. The FRAP region F has a surface area of $A=38\;\mu m^{2}≃\pi \;{\left(3.5\;\mu m\right)}^{2}$ in the 2D projection plane. The table provides the relevant properties of the reconstructed ER geometries adopted in the present study. Details on the reconstruction and representation of the surface meshes are provided in [45].

**Table 1 ijms-25-02993-t001:** A comparison of estimated quench rates and diffusion coefficients for the cell lines introduced in the left and right panels of Figure 2. The first row shows the results of Ref. [45]. The other two rows were obtained by two different approaches to the data analysis that are discussed in Section 3.1 and Section 3.2 of the present study. Their respective results are summarized in Figure 6 and Figure 7.

		Cell Line 1		Cell Line 2	
	Quantity	$r_{p}$	D	$r_{p}$	D
	Unit	s^−1^	μm^2^/s	s^−1^	μm^2^/s
Method					
Ref. [45]		0.0011	0.033	0.0016	0.0077
Equation (Equation 9)		—	0.05	—	0.015
Equation (Equation 11)		0.004 (1)	0.075 (15)	0.004 (1)	0.034 (13)

## Data Availability

Data is contained within the article.

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
