# Peer review of "Efficient Estimates of Surface Diffusion Parameters for Spatio-Temporally Resolved Virus Replication Dynamics"

_ijms, 2024, doi:10.3390/ijms25052993_

Round 1
Reviewer 1 Report
Comments and Suggestions for Authors
This work addresses computational study carried out in the tight connection with real experimental investigation of the transport process of fluorescent markers on a complex cell surface. The approach of fluorescent quenching is a modern method, which allows for obtaining highly accurate data. The latter form a basis for model studies aimed at revealing transport mechanisms in such a complex environment as the surface of biological objects. The simulation model built in this work is adjusted to the properties of a real object consists of one of the strongest points of this study. The application of the Laplace-Beltrami operator gives a realistic reproduction of displacements on curved surfaces, and the comparison with experimental data supports the validity of the model results. The scaling analysis presented in the second half of the manuscript is of high interest since it reveals universality features, which are a strong point of interest in the physics and biophysics of complex systems.
Thus, the content of this manuscript can be potentially of high interest to a wide auditory of specialists in molecular science, physics and biophysics, I recommend its acceptance.
In principle, I have one minor question, which is optional. Would it be possible to estimate and illustrate directly the dependence of the mean-squared displacement, i.e. the second moment of the spatial fluorescence distribution (in the Euclidean space) as a function of time? Although the FRAP signal intensity is a kind of proxy for this purpose, the explicit <R^2>=Dt^{\alpha} representation is more familiar to a wider auditory of physicists dealing with transport processes. Since diffusion in porous, tortuous systems and systems with traps and dead ends attracts active recent attention in physics and biophysics, the mentioned information may increase the impact of this work.
Author Response
Please find attached the pdf document, where the point to point analysis is performed.
We thank the Reviewer.

Reviewer 2 Report
Comments and Suggestions for Authors
The manuscript entitled "Efficient estimates of surface diffusion parameters for spatio-temporal resolved virus replication dynamics" elaborated by authors Markus M. Knodel, Gabriel Wittum and Jürgen Vollmer is a valuable foray into 3D spatio-temporal mathematical patterns. These approaches are very useful to perform advanced methods of treatment of diseases and also to elaborate strategies to fight against to virus invasion and to avoid the spread of epidemics. More precisely, this study deals the dynamics of intracellular virus replication, namely the replication of the Hepatitis C virus (HCV).
The paper is well written, having an appropriate structure in concordance with the scientific requirements of this multidisciplinary applied research domain. More exactly, the introductory section is clear, comprehensive and enlightening. Also, the presentation of the biological problem, the experimental measurements, and the numerical methods together with the scaling analysis of the model equations are well done. Finally, in the conclusions section the obtained results are well discussed and highlighted.
For all these reasons, I strongly recommend the publication of this very good work.
Author Response
We thank the Reviewer for his/her interest in our study and for the highly motivating evaluation / judgement of our results.
In particular, we thank the Reviewer for acknowledging and recommending our study for publication.